Exploring the value of hybrid capture-based next-generation sequencing technology in the suspected diagnosis of bloodstream infections

Liu Xinyuan 1
Gan Zhitao 2
Lin Zengshun 3
Lin Xiaojun 4
Yuan Jianying 3
Rong Lili 5
Chen Jiachang 3
Liu Jun 3
Li Yingzhen yingzhenli@foxmail.com 3
Hu Chaohui 1692673759@qq.com 1
1 KingMed School of Laboratory Medicine, Guangzhou Medical University , Guangzhou , Guangdong , China
2 Respiratory and Critical Care Medicine Department, Jinshazhou Hospital of Guangzhou University of Chinese Medicine , Guangzhou , Guangdong , China
3 Guangzhou KingCreate Biotechnologies Co., Ltd. , Guangzhou , Guangdong , China
4 Intensive Care Department, Jinshazhou Hospital of Guangzhou University of Chinese Medicine , Guangzhou , Guangdong , China
5 Clinical Laboratory, Jinshazhou Hospital of Guangzhou University of Chinese Medicine , Guangzhou , Guangdong , China
Braga Erika
Electronic publication date: 2024 Nov 8
Publication date: 2024
Volume: 12
Electronic Location ID: e18471
Received 2024 Apr 11; Accepted 2024 Oct 15
Copyright: ©2024 Liu et al.
Copyright year: 2024
Copyright holder: Liu et al.
License: This is an open access article distributed under the terms of the Creative Commons Attribution License, which permits unrestricted use, distribution, reproduction and adaptation in any medium and for any purpose provided that it is properly attributed. For attribution, the original author(s), title, publication source (PeerJ) and either DOI or URL of the article must be cited.
License URL: https://creativecommons.org/licenses/by/4.0/

Keywords: Bloodstream infections, Metagenomic next generation sequencing (mNGS), Blood culture, Clinical, Hybrid capture-based next generation sequencing

Funding: The authors received no funding for this work.

==============================
Background

Determining the source of infection is significant for the treatment of bloodstream infections (BSI). The gold standard of blood infection detection, blood cultures, have low positive rates to meet clinical needs. In this study, we investigated the ability of hybrid capture-based next generation sequencing technology to detect pathogens in peripheral blood samples collected from patients with suspected BSI. Blood cultures and capture sequencing assays were also analyzed against the final clinical diagnoses.

Methods

In this study, peripheral blood samples were collected from patients with fever, chills, and suspected BSI at Jinshazhou Hospital of Guangzhou University of Chinese Medicine from March 2023 to January 2024. All samples were tested by three different technologies: plasma capture sequencing technology, white blood cell capture sequencing technology, and blood culture. Relevant clinical diagnostic information was also collected. The performances of the blood cultures were then compared to those of both plasma capture sequencing technology and white blood cell capture sequencing technology.

Results

A total of 98 patients were included in this study. The positive rates of probe capture next generation sequencing (NGS technology) in plasma and white blood cells were 81.63% and 65.31%, respectively, which were both significantly higher than that of the blood culture, which was 21.43% (p < 0.001). Taking blood culture as the standard control, the sensitivity and specificity of plasma capture sequencing were 85.71% and 71.43%, respectively, while the sensitivity and specificity of white blood cell sequencing were 76.19% and 81.82%, respectively. Upon final clinical diagnosis, the clinical agreement rates of the blood cultures, plasma capture sequencing, and white blood cell capture sequencing were 39.80%, 83.67%, and 73.47%, respectively.

Conclusion

Our study demonstrates the high accuracy of probe capture sequencing technology compared to blood cultures in the identification of pathogenic microorganisms in BSI upon final clinical diagnosis. Among the different sample types, white blood cell samples had a lower clinical compliance rate compared to plasma samples, possibly due to the higher host rate in cell samples, which impairs the sensitivity of pathogen detection.

Introduction

Bloodstream infection (BSI) occurs when the bloodstream is invaded by a variety of pathogenic microorganisms (e.g., bacteria, fungi, viruses), leading to multiple clinical syndromes (Singer et al., 2016) with acute symptoms and rapid progression. The incidence of BSI ranges from 150 to 652 cases per 100,000 people (Cui et al., 2022; Kontula et al., 2021). Approximately 11 million sepsis-related deaths are reported worldwide each year and sepsis-related deaths accounted for 19.7% of all global deaths in 2017 (Rudd et al., 2020). The incidence and mortality of BSI increased during the COVID-19 pandemic (Taddei et al., 2023). When treating patients with critical infections, physicians typically use empiric broad-spectrum anti-infective therapy based on the condition of the patients (Timsit et al., 2020). However, this practice increases the risk of drug resistance and patient mortality (Rhee et al., 2020; Gradel et al., 2017) while prolonging the overall treatment cycle. Therefore, it is crucial to identify the causative pathogens as early as possible so the correct antibiotics can be used to improve therapeutic efficacy and reduce potential risks. Currently, the main technologies used in laboratories to detect pathogenic microorganisms in BSI are blood culture, smear microscopy, fluorescence in situ hybridization, and polymerase chain reaction (PCR) techniques (Edmiston et al., 2018). Nevertheless, the reliance on traditional technological tests to identify pathogen is often limited. Blood culture, currently the accepted gold standard for the diagnosis of BSI (Lamy, Sundqvist & Idelevich, 2020), sometimes proves incompetent due to its low positive rate, long turn-around time (TAT), and susceptibility to previous antibiotic use. Metagenomic next generation sequencing (mNGS) technology is widely used in clinical laboratory diagnostics, and hybrid capture-based next generation sequencing (NGS) technology (Gaudin & Desnues, 2018) is an emerging new method that can simultaneously detect DNA and RNA in patient samples. However, virtually no study has been conducted to explore the performance of such technology. Unlike previous multiplex PCR-based NGS technology (Huang et al., 2023) used in the clinic, this technique can detect tens of thousands of pathogens, including all common clinical pathogens and known antimicrobial resistance genes and virulence factors. In the testing process, blood samples are first centrifuged at different speeds to separate the plasma and then nucleic acids are extracted from the upper plasma layer, which contains mainly cfDNA. At the same time, there are a large number of white blood cells (WBC) between the plasma layer and blood cell layer, most pathogens are centrifuged to the blood cell layer, and some WBC also contain many phagocytized pathogens. Therefore, when analyzing the blood samples of patients with BSI, the difference between the detection of the hybrid capture sequencing method and the detection of the plasma layer and the WBC layer still needs to be further explored.

Materials and technology

In this study, we will explore the ability of capture NGS to detect pathogens such as bacteria, fungi and viruses in bloodstream infections. The positive rates between plasma and WBC will also be compared in the hope of providing accurate recommendations in sample selection for clinical testing of this technology. Comprehensive clinical diagnostic results of blood culture, blood routine, and antigen-antibody test will be taken as standards. The study plan is shown in Fig. 1.

Figure 1 The schematic of the study profile.

Blood drawn from suspected BSI patients was detected with blood culture and NGS simultaneously. After plasma centrifugation, the plasma and leukocyte layer were separated, and nucleic acid was extracted from each. The library was prepared for the capture sequencing, and the sequencing report was given after biological information analysis. Clinical diagnosis was given by the clinical doctor after considering any parameters and laboratory examination results. The performance of this methodology was compared with the hospital discharge report as the “gold standard”. Image credit: http://www.biorender.com, CC-BY-NC-ND license.

Patient collection

We selected 98 hospitalized patients with suspected BSI admitted to Jinshazhou Hospital of Guangzhou University of Chinese Medicine between May 2023 and January 2024. The study was approved by the Ethics Committee of Jinshazhou Hospital of Guangzhou University of Chinese Medicine, Guangzhou, China (Approval No. JSZ-IEC-SL-KT-20230403 (1)), and informed consent was taken from all the patients.

Case inclusion and exclusion criteria

Inclusion criteria were (1) patients with body temperature ≥38 °C or ≤36 °C, and at least one of the following standards: (2) (A) heart rate >90 beats/min, (B) respiratory rate >20 breaths/min or arterial blood carbon dioxide partial pressure PCO2 <32 mmHg, (C) recurrent chills, (D) coma or systemic symptoms of toxicity, or (E) skin and mucous membrane hemorrhage.

Exclusion criteria: (1) substandard sample volume (less than 3 mL) or (2) sample contamination from leakage due to improper storage.

Blood cultures

We collected 8–10 mL blood samples from patients with high fever and/or chills using aseptic blood collection technology and injected them into special blood culture bottles (one aerobic and one anaerobic bottle), shook well, and sent for examination within 2 h. According to the procedure of bacteria and fungi culture in the microbiology laboratory of Jinshazhou Hospital of Guangzhou University of Chinese Medicine, the blood samples were cultured by a BacT-ALERT 3D 240 automatic blood culturing system (bioMérieux, Marcy-l’Étoile, France) for 5 d. The strains were identified using the VITEK 2 Compact 60 (bioMérieux, Marcy-l’Étoile, France).

Capture sequencing

About 3–10 mL of blood was collected simultaneously from the above patients in disposable free DNA anticoagulation tubes (631010202P2111, Shandong Ande Healthcare Apparatus Co., Ltd., Zibo City, China). The samples were stored at 4 °C after collection and transported to Guangzhou KingCreate Biotechnology Co. for testing within 24 h. We then transferred 1.5 mL of whole blood to a new sterile EP tube, which was then centrifuged at 1,600× g for 10 min in 4 °C, and 300 µL of plasma was transferred for plasma nucleic acid extraction (KS121-WSWTQ-48, Guangzhou KingCreate Biotechnology Co., Ltd., Guangzhou, Guangdong, China). The remaining plasma was stored at −80 °C in new sterile tubes for spare use. After plasma removal, 100 µL leukocyte layer diluted in 500 µL saline was used for nucleic acid extraction with the same kit used in plasma nucleic acid extraction, and the remaining cells were stored at −80  °C for spare use. The walls of 600 µL diluted WBC were broken with 0.5 g glass grinding beads in ASD1000 cell crushers (ABclonal Technology, Wuhan, China) at 4,500 RPM for 45 s with 20 s intervals, repeated three times, which ensured that the nucleic acids released from difficult-to-labile organisms. The solution was then enriched by centrifugation at 12,000× g for 3 min, and 300 µL of supernatant was transferred to a new EP tube, with subsequent steps consistent with plasma extraction for DNA/RNA co-extraction. Finally, the solution was eluted with 50 µL nuclease free water. After nucleic acid extraction, the concentration of nucleic acids was determined and recorded using the Qubit® dsDNA HS and RNA Assay Kit instructions.

The extracted RNA was first reverse-transcribed using the Reverse Transcript Kit (KS630-NZL-48, Guangzhou KingCreate Biotechnology Co., Ltd.) to form cDNA. Then, the cDNA was applied to Library Construction Kit (KS130-CAPJK-48, Guangzhou KingCreate Biotechnology Co., Ltd., Guangzhou, Guangdong, China) to obtain a DNA library. Constructed libraries were then captured with a MetaCAP Hybridization-capture kit (KS129-CAPBH-6, Guangzhou KingCreate Biotechnology Co., Ltd., Guangzhou, Guangdong, China) (Cai et al., 2024). The captured libraries were applied to the MiniSeq platform (Illumina, USA) for 100-cycle sequencing, and the average sequencing data volume of each library was 1M reads. The data were analyzed in the bioinformatics analysis system and total data volume, Q30, and capture efficiency were used as quality control (PDseq-mNGS, Guangzhou KingCreate Biotechnology Co., Ltd., Guangzhou, Guangdong, China), and the microorganisms in each sample were reported. Each round of sequencing included a negative control (NC; plasma-free nucleic acids and a fragmented human genomic DNA mixture) and a positive control (a mixture of inactivated bacteria, fungi, and pseudovirus particles) (Shi et al., 2024). Bioinformatics analyses followed previous studies in which the group was involved (Cai et al., 2024).

Collection of clinical data

The following data of the included patients were collected from Jinshazhou Hospital of Guangzhou University of Chinese Medicine: gender, age, and results of blood routine examination including procalcitonin (PCT) and C-reactive protein (CRP). The results of the biochemical and immune test included a (1-3)-beta-D-glucan test and galactomannan test. The time to a positive result, result of sample blood culture, result of sequenced microbiological sequencing, and diagnostic result of hospital discharge were also collected.

Diagnostic performance analysis

The diagnostic results at the time of discharge of all subjects were used as the “gold standard”. The final diagnosis was fully discussed in the department in which all subjects were discharged from the hospital, and a retrospective final diagnosis was made by taking into account the patient’s clinical manifestations, various test results (including laboratory test and sequencing results), and treatment effects. The sensitivity, specificity, positive predictive value (PPV), negative predictive value (NPV), and overall compliance rate were calculated using the final clinical diagnosis as the reference standard.

Statistical analysis

Data were organized and processed using Microsoft Excel 2019, and statistical analysis was performed with SPSS 25.0 software. Counted data were described by the number of cases (percentage), and the Kolmogorov–Smirnov test was used to verify the normality of the data, which was described by the mean and standard deviation (SD) when the data showed a normal distribution. Non-normally distributed data were expressed as the median and interquartile range (IQR, P25, P75). A chi-square test was used to compare the specificity and sensitivity of blood culture, plasma capture sequencing, WBC capture sequencing, and final clinical diagnosis. Non-parametric tests analyzed using the Mann–Whitney U test for two independent samples from the same population, and statistical significance was set at P < 0.05.

Results

Clinical characteristics of patients

A total of 98 samples were collected from 56 males and 42 females, aged from 6 to 99 years with median age of 63.00 (43.00–73.00) years. The general data of the 98 patients are shown in Table 1. The medians for WBC, PCT, and CRP were 9.54, 3.41, and 59.62, respectively. It is worth noting that of all 98 patients, 21 patients with positive blood cultures took 26.00 (18.15, 31.50) hours to report results, and 77 patients with negative blood cultures took 5 days, while sequencing technology took only 18.40 (18.30,18.50) hours for 98 patients, which was a significantly shorter TAT compared to that when using blood culture technology (p < 0.05).

Table 1 Clinical characteristics of chosen suspected BSI patients.

	Patients with suspected bloodstream infections n = 98	
Median Age, (years)	63.00 (43.00–73.00)	
Male, n (%)	56 (57.14%)	
Indicators Laboratory examination		
White blood count, median (IQR) ×109/L	9.54 (6.22–14.77)	
Red blood cell count, median (IQR) ×1012/L	3.63 (2.71–4.27)	
Hemoglobin (g/L), median (IQR)	102.50 (76.75–125.50)	
Platelet count, median (IQR) ×109/L	190.50 (104.00–264.00)	
Neutrophil percentage %	80.45 (65.45–90.63)	
Lymphocytes percentage %	11.70 (4.70–18.80)	
Procalcitonin (µg/L), median (IQR)	3.41 (1.61–7.71)	
Interleukin-6 (pg/mL), median (IQR)	85.95 (14.77–225.98)	
C-reactive protein (mg/L), median (IQR)	59.62 (11.64–147.55)	
else		
Time-consuming to report positive blood cultures (h)	26.00 (18.15–31.50)	
Total time for probe capture sequencing (h)	18.40 (18.30–18.50)	

Comparison of capture sequencing and blood culture for pathogen detection

After comparing the detection rates of the two technologies, hybrid capture-based sequencing showed a higher detection rate of pathogenic microorganisms compared to that of blood culture. The number of patients with positive blood culture, plasma capture sequencing, and WBC capture sequencing was 21 (21.43%), 80 (81.63%), and 64 (65.31%), respectively. The blood culture’s positive rate was significantly lower than that of both capture sequencing technologies, and the difference was statistically significant (p < 0.001, both). Among all positive samples, 40 were detected with bacteria or fungi, and only 40 positive samples were detected with virus using plasma capture sequencing. In WBC capture sequencing, 30 samples were detected with bacteria or fungi, and 34 samples were detected with only viruses. Bacteria and fungi were detected in 21 samples of positive blood culture. The specific distribution is described in Fig. 2.

Figure 2 Description of capture sequencing and blood culture results of suspected BSI patients.

(A) Negative and positive rate of capture sequencing and blood culture in suspected BSI patients. (B) Types of pathogens diagnosed in each sample under different testing techniques.

The results showed that the top three of the 13 bacteria and two fungi detected in blood cultures were Klebsiella pneumoniae (N = 8), Pseudomonas aeruginosa (N = 3), and Staphylococcus hominis (N = 3). Of the 18 bacteria and eight fungi detected by plasma capture sequencing, the top three were K.pneumoniae (N = 15), P. aeruginosa (N = 10), and Acinetobacter baumannii (N = 4). Of the 14 bacteria and five fungi detected by WBC capture sequencing, the top three were K. pneumoniae (N = 11), P. aeruginosa (N = 8), and A. baumannii (N = 3). Compared to blood culture, capture sequencing could detect more pathogens (plasma: 176 vs. 28, p < 0.001; WBC: 131 vs. 28, p < 0.001). In addition, 11 cases of fungi detection in plasma capture sequencing included five cases of Candida, four cases of Aspergillus, and one case of Pneumocystis. The five cases of fungi detected by WBC capture sequencing included two cases of Candida, two cases of Aspergillus, and one case of Pneumocystis. Only two cases of Candida were detected using blood culture technology. The most common viruses detected in both types of capture sequencing were Human gammaherpesvirus 4 (EBV), Human betaherpesvirus 5 (CMV) and Torque teno virus (TTV), which compensates for the inability of blood cultures to detect viruses. In addition, one case of RNA virus (Dengue virus 1) was detected in both plasma and WBC capture sequencing with consistent results. The specific distribution is shown in Fig. 3.

Figure 3 Distribution of detected pathogenic microorganism of capture sequencing and blood culture in suspected BSI patients.

For bacteria and fungi results, the detection rate of plasma capture sequencing was higher than WBC capture sequencing and blood culture (40.81% > 30.61% > 21.43%), as presented in Table 2. The differences were statistically significant for blood culture versus plasma capture sequencing (p < 0.05). When considering only bacteria and fungi and using blood culture as the standard, plasma capture sequencing was slightly more sensitive (85.71%) than WBC capture sequencing (76.19%), with specificity of 71.43% and 81.82%, respectively, when compared to blood culture. These findings are summarized in Table 3.

When using final clinical diagnosis as the standard, 80 of the 98 total samples were infection-positive and 18 were non-infected. Among the 80 clinically diagnosed infection-positive samples, 21 were positive on blood culture, 72 were positive on plasma capture sequencing, and 59 were positive on WBC capture sequencing. All 18 clinically diagnosed non-infection samples were culture-negative, and there were 10 and 13 plasma and WBC capture sequencing-negative cases, respectively. The overall compliance rate was 39.80% for blood cultures and 83.67% and 73.47% for plasma and WBC capture sequencing, respectively (Table 4).

Result comparison between plasma and WBC capture sequencing groups

In order to investigate the detection performance of probe capture sequencing in blood samples, we continued to compare the efficiency of plasma capture sequencing and WBC capture sequencing, and classified them by microbial types. Additionally, in the detected bacteria, viruses, fungi, and all pathogens, the median of the ratio of pathogen sequence counts detected by plasma capture sequencing to those detected by leukocyte capture sequencing were 1.68 (1.42, 14.82), 2.08 (0.83, 6.42), 1.71 (0.60, 3.17), and 1.94 (0.93, 6.42), respectively. The detection performance of the plasma group in the overall detection was significantly higher than that of the WBC group (p < 0.001).

Table 2 Positive rate comparison between blood capture sequencing and blood culture detection (only fungi and bacteria detection were considered).

	Blood culture	plasma capture sequencing	WBC capture sequencing	
Positive	21 (21.43%)	40 (40.81%)	30 (30.61%)	
Negative	77	58	68	

Table 3 Diagnostic performance of blood capture sequencing in suspected BSI patients, compared with blood culture.

	Blood culture
positive	Blood culture
negative	Sensitivity
(%)	Specificity
(%)	PPV
(%)	NPV
(%)	Total consistent rate (%)	
Plasma capture sequencing positive	18	22	85.71	71.43	45.00	94.83	74.49	
Plasma capture sequencing negative	3	55						
WBC capture sequencing positive	16	14	76.19	81.82	53.33	92.65	80.61	
WBC capture sequencing negative	5	63						

Table 4 Diagnostic performance comparison between blood capture sequencing and blood culture for suspected BSI patients.

	Clinical final
diagnosis positive	Clinical final
diagnosis negative	Sensitivity
(%)	Specificity
(%)	PPV
(%)	NPV
(%)	Total consistent
rate (%)	
Blood culture positive	21	0	26.25	100	100	23.38	39.80	
Blood culture negative	59	18						
Plasma capture sequencing positive	72	8	90.00	55.56	90	55.56	83.67	
Plasma capture sequencing negative	8	10						
WBC sequencing positive	59	5	73.75	72.22	92.19	38.24	73.47	
WBC sequencing negative	21	13						

We also analyzed the consistency of pathogens identified by blood sequencing in all patients. If the pathogens identified in both fractions were identical, the test results were considered a match. If there was at least one pathogen overlap in pathogen detection, the results were considered partially matched. If the pathogens identified by the two methods were completely different, the results were considered a mismatch. In this study, 53.09% of the pathogens identified were a match, 20.99% were partially matched, and 25.93% were completely mismatched. Among the mismatched results, 80.95% were plasma capture sequencing positive, 4.76% were WBC capture sequencing positive, and 14.29% were both mismatches (Fig. 4).

Figure 4 Comparison of plasma capture sequencing and WBC capture sequencing results.

(A) RPM ratio of detected pathogenic microorganism between plasma capture sequencing and WBC capture sequencing (RPM-plasma/RPM-WBC), classified with the type of pathogen. Data was shown in the level of median. (B) The pie chart showed the matching distribution of plasma capture sequencing and WBC capture sequencing. The mismatch samples were classified as plasma-positive, WBC-positive, and both mismatched. (C) Comparison of RPM of all detected pathogens in plasma capture sequencing and WBC capture sequencing.

Discussion

The prevalence of BSI in hospitalized patients is associated with multiple factors, including aging, an increasing rate of underlying diseases and invasive surgical diseases, abuse of broad-spectrum antibiotics, and the overuse of immuno-suppressants (Weng et al., 2023; Padro et al., 2019; Tsuzuki et al., 2021). Clinicians typically turn to blood culture for pathogenic detection, which generally provides detailed information of the pathogens and their resistance conditions within 2-6 days. Some may take a longer time, up to several weeks, since pathogens such as fungi and mycobacteria grow so slowly; some may have false-negative results because the pathogen simply cannot be cultured. Some detection technologies using pathogens or host biomarkers are also used to aid diagnosis, such as (1,3)-β-D-glucan (G), galactomannan (GM), C-reactive protein, and procalcitoninogen (Povoa et al., 2023; Song et al., 2019), yet most technology have limited sensitivity or specificity compared to mNGS technology. A lack of general biomarkers also requires more co-diagnostic technology simultaneously conducted for final conclusions. As timely antibiotic therapy provides significant advantages in BSI treatments (Rhodes et al., 2017; Niederman et al., 2021), inappropriate empirical antibiotic abuse can lead to increased mortality (Kang et al., 2005), and researchers have devoted considerable efforts to develop a simple, sensitive, and broad-spectrum pathogen detection technology. PCR-based technologies, such as digital PCR, are highly sensitive but have a narrow detection range (Wu et al., 2022). DNA melting curve technology can further expand the detection range of digital PCR and real time PCR to dozens of species (Traylor et al., 2024; Goshia et al., 2024). Sequencing-based technology such as Sanger Sequencing, mNGS, and nanopore sequencing, have undergone rapid development in recent years. Thus, the hybrid capture-based next generation sequencing technology that can detect both DNA and RNA pathogens in bloodstream sample provides promising detection in future BSI diagnosis.

Our capture sequencing demonstrates better in relation to detection performance. A total of 98 patients with suspected BSI were included in this study. The three most frequently detected pathogens by hybrid capture-based next generation sequencing technology were K. pneumoniae, P. aeruginosa, and A. baumannii, with all common pathogens in line with previous study of hospital-associated infections (Sinto et al., 2022). The results of hybrid capture-based next generation sequencing technology were inconsistent with the results of blood culture (p < 0.001). The positive rates of probe capture sequencing technology in plasma samples and WBC samples were 81.63% and 65.31%, respectively, which were significantly higher than that of blood culture technology (21.43%). In Table 5, we present the results of comparing mNGS technology with blood culture in different types of BSI patients (Geng et al., 2021; Liu et al., 2023; Sun et al., 2022; Zhang et al., 2022; Zuo et al., 2023; Schulz et al., 2022; Zhou et al., 2023; Lee et al., 2022). Our blood culture positivity rate was mostly consistent with them, but the positivity rate of hybridization capture sequencing was significantly higher than the results of (Liu et al., 2023) (68.5% vs. 26.5%), (Sun et al., 2022) (67.74% vs. 19.35%), and (Zuo et al., 2023) (50.54% vs. 23.04%). Even when compared with the results of confirmed sepsis patients, our detection rate was still the highest, suggesting that the hybrid capture-based sequencing technology has better sensitivity in the detection of pathogens in blood samples. According to our clinical diagnosis results, the clinical coincidence rates of hybrid capture-based next generation sequencing technology (plasma 83.67%/WBC 73.47%) were significantly higher than those of the blood culture (39.80%), (p < 0.001, both). The detection performance of hybrid sequencing technology has been greatly improved compared to other technologies, making it more suitable for clinicians to use this tool to accurately diagnose and adjust medication for patients.

Table 5 Detection of mNGS technique in the BSI population.

Previous studies related to BSI	Target patients (n)	Sequencing positive (%)	Blood culture positive (%)	Refs	
Our study	Suspected BSI (98)	Capture-seq = 81.63%, (plasma)
Capture-seq = 65.31%, (WBC)	BC = 21.43%		
1	Suspected BSI (63)	mNGS = 41.3%, (plasma)	BC = 7.9%	Geng et al. (2021)	
2	Suspected BSI (162)	mNGS = 68.5%, (plasma)	BC = 26.5%	Liu et al. (2023)	
3	Spesis (124)	mNGS = 67.74%, (plasma)	BC = 19.35%	Sun et al. (2022)	
4	Identified or suspected bacteremia (114)	mNGS = 56.14%, (plasma)	BC = 43.86%	Zhang et al. (2022)	
5	Suspected Sepsis (277)	mNGS = 50.54%, (plasma)	BC = 23.04%	Zuo et al. (2023)	
6	Suspected BSI (97)	mNGS = 43%, (plasma)	BC = 14%	Schulz et al. (2022)	
7	Suspected BSI (99)	mNGS = 65.66%, (plasma)	BC = 13.13%	Zhou et al. (2023)	
8	Spesis (48)	mNGS = 62.5%, (plasma)	BC = 14.5%	Lee et al. (2022)	

The detection performance of hybrid capture sequencing in plasma is superior to that of WBC. Blood culture only detects culturable bacteria and fungi, but hybrid capture-based next generation sequencing technology can detect viruses, parasites, and unculturable bacteria/fungi. Generally, nucleic acids in plasma are mostly cfDNA in humans. In the early stages of infection, the host’s immune responses are not yet triggered, and the nucleic acids of pathogens have therefore not been released into the plasma. This could explain the inconsistency of blood culture versus hybrid capture-based next generation sequencing technology in plasma. In the leukocyte layer, some pathogens are deposited after centrifugation, and the pathogenic microorganisms are ingested by leukocytes. Nevertheless, as the largest portion of the extracted nucleic acids were from humans, probe capture sequencing technology in the blood cell also returned negative results. Of all the pathogens, the RPM median level of the plasma group reached 1.94 (0.93, 6.42) compared to the median of the WBC group. Taking blood culture as the control, hybrid capture-based next generation sequencing technology was more sensitive in plasma than in WBC samples (p < 0.001). Such results are inconsistent with the conclusions from the study conducted by Wu et al. (2023), suggesting that if the de-host step is not performed, plasma layers are more recommended for pathogen detection.

Capture sequencing has a wider detection range and shorter time consumption compared to culture methods. In this study, we found that the overall TAT of blood culture was approximately 26 h, whereas hybrid capture-based next generation sequencing technology released final results in about 18 h, which was even shorter than regular mNGS technology which required 24–48 h, and the increase in absolute mortality associated with an hour’s delay in antibiotic administration was 0.3% (Liu et al., 2017). Shorter TAT provides clinicians with timely and professional results to guide further treatments in BSI. There were also some special cases in this study that deserve more in-depth discussion. For example, one case tested positive for dengue virus by capture sequencing, and that patient tested equally positive by immunological testing, with consistent results. Cases with positive results of RNA viruses and parasites in this study were very limited, possibly due to the limitation of where samples were collected. Future studies with larger sample sizes are needed to improve our data volume and data diversity.

There were, however, several limitations in this study. First, this was an observational and non-interventional clinical study, which means this study cannot evaluate the impact of hybrid capture-based next generation sequencing technology on clinical treatments. Second, inconsistent results from different technologies were not validated by a third technology such as ddPCR. Third, due to limitations of time and cost, blood cells were not de-hosted, meaning the test results of blood cell samples were not at their most optimal. Fourth,some patients received antibiotic treatments prior to blood collection, which may have affected the positive rates and thus the sensitivity of the final results.

As for the hybrid capture-based next generation sequencing technology, it demonstrates excellent performance in clinical blood infection diagnosis. However, there are still some issues to be optimized: (1) the operational procedure is more complicated than regular mNGS and requires better maneuverability from the conductors; (2) it is difficult to determine the real “smoking gun” when multiple microbes are detected from one sample; and (3) the capability of detecting emerging pathogens still need to confirm. Nevertheless, the hybrid capture-based next generation sequencing technology is still being optimized to overcome these shortcomings and is expected to be widely applied to the diagnosis of bloodstream infections in near future.

Conclusion

Hybrid capture-based NGS technology can detect multiple pathogens in one sample including bacteria, fungi, viruses, and parasites. In the cases of suspected bloodstream infections, positive rates and clinical coincidence rates of hybrid capture-based next generation sequencing technology are significantly higher than blood culture technology, particularly in plasma samples. Hybrid capture-based next generation sequencing technology proves to be a promising technology in clinical BSI diagnosis, as other regular BSI detection technology have lower sensitivity.

Supplemental Information

Supplemental Information 1 Raw data

We thank Ms. Hongxia Luo, Mr. Hailing Wanghu, and Ms. Shangqing Chen for proofreading and editing this article. Thanks to Mr. Kangze Pu and Ms. Fengming Guo for finishing the experimental data of this research. We appreciate linguistic assistance provided by Mr. Yunxiao Liu and Mr. Jiajun Mo during the preparation of this manuscript.

Additional Information and Declarations

Competing Interests

Author Contributions

Human Ethics

Data Availability

Zengshun Lin, Jianying Yuan, Jiachang Chen, Jun Liu and Yingzhen Li are employed by Guangzhou KingCreate Biotechnologies Co., Ltd.

Xinyuan Liu conceived and designed the experiments, performed the experiments, analyzed the data, prepared figures and/or tables, authored or reviewed drafts of the article, and approved the final draft.

Zhitao Gan conceived and designed the experiments, performed the experiments, analyzed the data, prepared figures and/or tables, authored or reviewed drafts of the article, and approved the final draft.

Zengshun Lin conceived and designed the experiments, performed the experiments, analyzed the data, prepared figures and/or tables, authored or reviewed drafts of the article, and approved the final draft.

Xiaojun Lin conceived and designed the experiments, performed the experiments, analyzed the data, prepared figures and/or tables, authored or reviewed drafts of the article, and approved the final draft.

Jianying Yuan conceived and designed the experiments, performed the experiments, analyzed the data, prepared figures and/or tables, authored or reviewed drafts of the article, and approved the final draft.

Lili Rong conceived and designed the experiments, performed the experiments, analyzed the data, prepared figures and/or tables, authored or reviewed drafts of the article, and approved the final draft.

Jiachang Chen conceived and designed the experiments, performed the experiments, analyzed the data, prepared figures and/or tables, authored or reviewed drafts of the article, and approved the final draft.

Jun Liu conceived and designed the experiments, performed the experiments, analyzed the data, prepared figures and/or tables, authored or reviewed drafts of the article, and approved the final draft.

Yingzhen Li conceived and designed the experiments, performed the experiments, analyzed the data, prepared figures and/or tables, authored or reviewed drafts of the article, and approved the final draft.

Chaohui Hu conceived and designed the experiments, performed the experiments, analyzed the data, prepared figures and/or tables, authored or reviewed drafts of the article, and approved the final draft.

The following information was supplied relating to ethical approvals (i.e., approving body and any reference numbers):

This research was approved by the Ethics Committee of Jinshazhou Hospital of Guangzhou University of Chinese Medicine, Guangzhou, China

The following information was supplied regarding data availability:

The sequence data is available at NCBI: PRJNA1096997.

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
