# Peer review of "Exploring the value of hybrid capture-based next-generation sequencing technology in the suspected diagnosis of bloodstream infections"

_PeerJ, doi:10.7717/peerj.18471_

## Round 0.1 · original submission · Major Revisions

The review process has been completed, and two highly qualified referees have provided thorough feedback, which you can find at the end of this letter. There are, however, some concerns that need to be addressed in your resubmission. I concur with the reviewers and emphasize the importance of meticulously addressing the points they raised and incorporating all necessary changes into the manuscript. Additionally, it is crucial to improve the English language to ensure clarity and comprehensibility for readers.

·

Basic reporting

This study compares detection rate of bloodstream infections (BSI) between hybrid capture-based next sequencing technology and gold standard blood culture technology. Both plasma and white-blood cells from each blood sample are processed with hybrid-capture based NGS. The authors found detection rate of hybrid-capture based NGS from plasma to be better compared to other methods. Publication’s introductory section provides essential context, ensuring readers understand the problem statement from outset.
Major issues with basic reporting are as follows:
1) The clarity of your text for an international audience would benefit from enhancing the English language. Notably, improvements are needed in several instances, such as lines 8-9, 71-73, 112-114, 124-125, 133-134, 187-188, 189-192 and 205-206, where the current wording poses challenges to comprehension.
2) The manuscript's main analysis relies on sequencing data, yet the raw data essential for reproducing these findings is unavailable. Underlying data related to its figures is also missing. These omissions limit the transparency and verifiability of the study's conclusions, hindering the ability of researchers to independently assess and build upon the reported results.
3) Code used to generate figures and execute the bioinformatics workflow for quantifying results from the sequencing datasets has not been provided, thereby impeding the reproducibility of the findings.
Minor issue with basic reporting is as follows:
1) While the manuscript presents figures relevant to its findings, some improvements are needed for better readability. Specifically, certain figures would benefit from adjustments in font sizes on axis labels and descriptions. For instance, Figure 3's Y-axis uses two different scaling factors, complicating comparisons across elements. Additionally, enhancing the font color and size on both axes of Figure 4B would improve overall readability.

Experimental design

Major issues with experimental design are as follows:
1) Lines 80 - 92 and lines 103 - 128: This study faces the challenge of distinguishing whether detection of specific bacteria in plasma is attributable to infection or circulating microbial cell-free DNA (cfmDNA), which can originate from bacteria in various body sites, including the gut. Therefore, it would be more appropriate to assess differential bacterial abundance between disease and control groups. The study included 98 patients suspected of bloodstream infections, among whom only a small subset (18) were found to be without infections. The limited number of control subjects reduces the reliability of differential analysis. Increasing the number of control subjects would enhance the study's specificity.
2) Lines 126 - 128: The authors utilized a bioinformatics analysis system (PDseq-mNGS, Guangzhou KingCreate Biotechnology Co., Ltd.) to analyze the sequencing data. To ensure reproducibility of the results, it is recommended to include detailed steps of the bioinformatics pipeline used in the analysis. This should encompass the specific algorithms, parameters, and software versions employed, facilitating transparency and reproducibility of the findings.
3) Lines 91 - 92 and lines 103 - 119:The methodologies described in these sections do not specify whether the tubes were assessed for plasma hemolysis. This information is crucial because hemolysis can introduce artifacts that affect the reliability of various downstream analyses. Hemolysis can release intracellular components into the plasma, potentially skewing results or interpretations.
4) Lines 103 - 128: Based on the methodologies described in this section, it is not evident whether negative controls were included on each sequencing plate. Incorporating negative controls is crucial as they help ensure that observations from the experiment and subsequent analyses are not influenced by contaminants. These controls serve to validate the absence of unintended microbial or environmental DNA that could otherwise skew the results or interpretations of the sequencing data. Therefore, it is recommended to clarify whether such controls were utilized to maintain the integrity and reliability of the study findings.
5) Lines 103 - 128: The methodologies in this section specify volumes for biosamples at various steps, but they lack rationale regarding the selection of these volumes. For instance, it is unclear whether there was consideration of theoretical limits of detection for the smallest size of bacteria in plasma, and if this influenced the choice of minimum plasma volume. Providing clarity on how these volumes were determined based on technical limitations or analytical thresholds would strengthen the methodology and enhance the reproducibility and interpretation of the study results.
6) Lines 293 - 294: Some patients included in the study had received antibiotic treatment before blood sample collection. Ideally, subjects who have undergone antibiotic treatment should be excluded, as this factor significantly complicates result interpretation and introduces confounding variables.

Minor issues with experimental design are as follows:
1) Lines 86 and 92 and 216 - 218: The authors note that the misuse of broad-spectrum antibiotics can complicate the detection of bloodstream infections (BSI). However, it is not specified whether a criterion such as "history of antibiotic misuse" was considered during subject selection for the study.
2) Lines 123 - 124: The MetaCAP Hybridization capture kit (KS129-CAPBH-64, Guangzhou KingCreate Biotechnology Co., Ltd.) was utilized for target selection. To enhance transparency and facilitate reproducibility, it would be beneficial to provide a reference or supplementary document that lists the probes included in this kit alongside the manuscript submission. This additional information would assist readers and researchers in understanding the specific targets captured during the study and enable them to replicate or build upon the findings more effectively.
3) Lines 120 - 128: Based on the methodologies outlined in this section, it remains uncertain whether any quality control measures or analyses were conducted after the sequencing process. Post-sequencing quality control is essential to assess the integrity and reliability of the generated data. These steps typically involve evaluating sequencing depth, assessing read quality, checking for adapter contamination, and confirming the absence of technical artifacts that could affect downstream analyses and interpretations. Including details about these quality control procedures would enhance transparency and ensure the robustness of the study's findings.

Validity of the findings

Major issues with the results section are as follows:
1) Lines 158 - 160: This section highlights that the turnaround time (TAT) for sequencing technology was significantly shorter than that for blood culture. The TAT was reported for 21 patients in the case of blood culture, whereas it was reported for 98 patients with sequencing technology. However, it remains unclear why blood culture was performed or reported for fewer patients than sequencing technology.
2) Lines 161 - 214: This section highlights that plasma sequencing technology demonstrates better detection rate compared to other methods. However, distinguishing between the origin of DNA: whether it signifies an infection or circulating microbial cell-free DNA (cfmDNA), poses a challenge. Consequently, it is difficult to definitively assess whether the observed higher detection rate in plasma truly reflects improved diagnostic accuracy for infections.
3) Lines 188 - 199: This section reports that sequencing technology-based methods demonstrate higher sensitivity compared to blood culture. However, the reported results suggest that blood cultures exhibit significantly better specificity. Achieving a low false positive rate is crucial, as indiscriminate use of antibiotics, as mentioned by the authors, can lead to adverse consequences. It is important to balance sensitivity with specificity in diagnostic testing to avoid unnecessary treatments and ensure accurate identification of true infections. Authors are encouraged to also elaborate on the lower specificity associated with sequencing technology.
4) Lines 154 - 214: The methods section indicates that the concentration of nucleic acids was determined and recorded using the Qubit® dsDNA HS Assay Kit. However, these results are not presented in the results, discussion sections, or supplementary documents. It would be beneficial to include these findings as they provide essential information regarding the quantity of nucleic acids extracted and can influence downstream analyses and interpretations.
5) Lines 154 - 214: The results section lacks any findings from quality control steps or analyses conducted during the study. Including these results is crucial as they provide insights into the reliability and integrity of the data generated.

Minor issue with results section is as follows:
1) Lines 158 - 160: For clarity, it would be beneficial to report the time taken for each step within the blood culture and sequencing technology workflows, rather than solely the overall turnaround time (TAT).

Additional comments

The manuscript exhibits significant issues that need to be addressed prior to publication. These include ambiguities in the English language throughout the text, absence of code for figures and analyses, crucial omissions in detailing both lab and bioinformatics workflow steps, insufficient consideration of confounding factors, and a lack of documentation on quality control procedures both in the laboratory and informatics processes. These deficiencies collectively hinder the clarity, reproducibility, and reliability of the findings presented.

Reviewer 2 ·

Basic reporting

1. Abstract/ Background section: Original text: “The gold standard of blood infection detection, blood culture, has a positive rate to low to meet the clinical needs.” Rephrase to “…low positive rate to meet clinical needs” instead. Or “positive rate too low to meet clinical needs”
2. Abstract/ Background section: Original text: “Blood cultures and biochemical technology are also conducted as test controls.”. Clarify what is the control here – final clinical diagnosis seems to be the control.
3. Abstract/ results: Please clarify that "clinical agreement rate" is against final diagnosis.
4. Abstract/ Conclusion: Original text: “high accuracy of probe capture sequencing technology” – high accuracy in comparison with what?
5. Introduction: “were even amounting” – please clarify the meaning here
6. Introduction: Correct typo: technologyological > technological
7. Materials and technology > patient collection : “signed an” - remove
8. Exclusion criteria: define substandard sample volume
9. Collection of clinical data: The time to positive and result of simple bacteria culture – please clarify this phrase. Is it time to positive result and result of simple bacteria culture?
10. Discussion: Original text: “some returns negative result” – some “return”
11. Discussion: Original text: “were Significant higher than the blood culture (39.80%)” – “significantly” instead
12. Discussion section: what is the de-host step? Elaborate.
13. Discussion section: “one case returned positive result of DengueVirus, which consistent” – which is consistent
14. Discussion section: “forth” > fourth

Experimental design

1. Statistical analysis: and Kolmogorov-Smirnov was used to verify the normality of the data - Which data is being referred to here? That is, sequencing result/ time to diagnosis - what parameter is being referred to?
2. Statistical analysis: And chi-square test was used to compare the measurements. - Which measurements are being compared?
3. Statistical analysis: Mann-Whitney U test was employed for comparing the differences in the continuous variables, --- What are the continuous variables in this context?

Validity of the findings

1. Comparison of capture sequencing and blood culture for pathogen detection: Original text: “Blood cultures were statistically significant when comparing the two sequencing technology (P<0.001, both)”. Clarify what (parameter) is statistically significant when comparing blood culture against these 2 technologies. i.e. in this case positivity rate? Is this considered to be a continuous variable or counted data? Justify the use of the statistical test applied.
2. the median of ratio - Is this detection rate of sequencing technology vs. blood culture?
3. Discussion section: Original text: “but the positivity rate of hybridization capture sequencing is significantly higher than the results of Hongxia Duan et al. (68.5% vs. 26.5%), Limin Sun et al. (67.74% vs. 19.35%),”. These studies seem to have had higher sample size, so their detection rate confidence intervals would be tighter than this study. Please comment.
4. Discussion section: Original text: “some patients received antibiotic treatments prior to blood collection, which may affect the positive rates and thus the sensitivity of the final results.” – why were these patients not excluded? There was no mention of this factor earlier in the paper.

Additional comments

Thank you for conducting a small but interesting study comparing blood culture, sequencing technology based pathogen detection. In the discussion section: Is it possible to compare / comment on DNA melt technology that has shorter TAT when compared to NGS for pathogen detection?

Annotated reviews are not available for download in order to protect the identity of reviewers who chose to remain anonymous.

---

## Round 0.2 · Minor Revisions

The manuscript has shown improvement, but it still requires further attention. Incorporating a comment regarding DNA melt curve technology in the discussion section is essential to ensuring a comprehensive analysis of the findings (see reviewer #2 comment).

Minor comments:
1- The scientific names already cited in full (genus and species) should have the genus abbreviated and the species name maintained in subsequent citations throughout the text. For example, Klebsiella pneumoniae = K. pneumoniae.
2- Lines 94-96: Better explain the novelty of the study. What kind of pathogens? Include some potential results to give the readers an idea of the study's importance.
3 - Lines 96-97: Transfer the information The study plan is 97 shown in Figure 1." to Methods Section.
4 -Line 269: "Our capture sequencing demonstrates better detection performance. " Complete the sentence "better in relation to..."

Reviewer 2 ·

Basic reporting

No additional comments

Experimental design

No additional comments

Validity of the findings

No additional comments

Additional comments

Need to include the comment about DNA melt curve technology in the discussions section

---

## Round 0.3 · accepted · Accept

The authors have addressed all of the comments.